# Hypothesis Testing in Unsupervised Domain Adaptation with Applications in Alzheimer's Disease

**Hao Henry Zhou**[†]     **Sathya N. Ravi**[†]     **Vamsi K. Ithapu**[†]
**Sterling C. Johnson**[§,†]     **Grace Wahba**[†]     **Vikas Singh**[†]

[§]William S. Middleton Memorial VA Hospital     [†]University of Wisconsin–Madison

## Abstract

Consider samples from two different data sources $\{\mathbf{x}_s^i\} \sim P_{\text{source}}$ and $\{\mathbf{x}_t^i\} \sim P_{\text{target}}$. We only observe their transformed versions $h(\mathbf{x}_s^i)$ and $g(\mathbf{x}_t^i)$, for some known function class $h(\cdot)$ and $g(\cdot)$. Our goal is to perform a statistical test checking if $P_{\text{source}} = P_{\text{target}}$ while removing the distortions induced by the transformations. This problem is closely related to domain adaptation, and in our case, is motivated by the need to combine clinical and imaging based biomarkers from multiple sites and/or batches – a fairly common impediment in conducting analyses with much larger sample sizes. We address this problem using ideas from hypothesis testing on the transformed measurements, wherein the distortions need to be estimated *in tandem* with the testing. We derive a simple algorithm and study its convergence and consistency properties in detail, and provide lower-bound strategies based on recent work in continuous optimization. On a dataset of individuals at risk for Alzheimer's disease, our framework is competitive with alternative procedures that are twice as expensive and in some cases operationally infeasible to implement.

## 1   Introduction

A first order requirement in many estimation tasks is that the training and testing samples are from the same underlying distribution and the associated features are directly comparable. But in many real world datasets, training/testing (or source/target) samples may come from different "domains": they may be variously represented and involve different marginal distributions [8, 32]. "Domain adaptation" (DA) algorithms [24, 27] are often used to address such problems. For example, in vision, not accounting for systematic source/target variations in images due to commodity versus professional camera equipment yields poor accuracy for visual recognition; here, these schemes can be used to match the source/target distributions or identify intermediate latent representations [12, 1, 9], often yielding superior performance [29, 12, 1, 9]. Such success has lead to specialized formulations, for instance, when target annotations are absent (unsupervised) [11, 13] or minimally available (semi-supervised) [7, 22]. With a mapping to compensate for this domain shift, we know that the normalized (or transformed) features are sufficiently invariant and reliable in practice.

In numerous DA applications, the interest is in seamlessly translating a *classifier* across domains — consequently, the model's test/target predictive performance serves the intended goals. However, in many areas of science, issues concerning the statistical power of the experiment, the sample sizes needed to achieve this power and whether we can derive $p$-values for the estimated domain adaptation model are equally, if not, more important. For instance, the differences in instrument calibration and reagents in wet lab experiments are potential DA applications except that the downstream analysis may involve little to no discrimination performance measures per se. Separately, in multi-site population studies [17, 18, 21], where due to operational reasons, recruitment and data acquisition is distributed over multiple sites (even countries) — site-specific *shifts* in measurements and missing covariates are common [17, 18, 21]. The need to *harmonize* such data requires some form of DA. While good

predictive performance is useful, the ability to perform hypothesis tests and obtain interpretable statistical quantities remain central to the conduct of experiments or analyses across a majority of scientific disciplines. We remark that constructs such as $\mathcal{H}\Delta\mathcal{H}$ distance have been widely used to analyze non-conservative DA and obtain probabilistic bounds on the performance of a classifier from certain hypotheses classes, but the statistical considerations identified above are not well studied and do not follow straightforwardly from the learning theoretic results derived in [2, 5].

**A Motivating Example from Neuroscience.** The social and financial burden (of health-care) is projected to grow considerably since elderly are the fastest growing populace [28, 6], and age is the strongest risk factor for neurological disorders such as Alzheimer's disease (AD). Although numerous large scale projects study the aging brain to identify early biomarkers for various types of dementia, when younger cohorts are analyzed (*farther* away from disease onset), the effect sizes become worse. This has led to multi-center research collaborations and clinical trials in an effort to increase sample sizes. Despite the promise, combining data across sites pose significant statistical challenges – for AD in particular, the need for harmonization or standardization (i.e., domain adaptation) was found to be essential [20, 34] in the analysis of multi-site Cerebrospinal fluid (CSF) assays and brain volumetric measurements. These analyses refer to the use of AD related pathological biomarkers ($\beta$-amyloid peptide in CSF), but there is variability in absolute concentrations due to CSF collection and storage procedures [34]. Similar variability issues exist for amyloid and structural brain imaging studies, and are impediments before multi-site data can be pooled and analyzed in totality. The temporary solution emerging from [20] is to use an "normalization/anchor" cohort of individuals which will then be validated using test/retest variation. The goal of this paper is to provide a rigorous statistical framework for addressing these challenges that will make domain adaptation an analysis tool in neuroimaging as well as other experimental areas.

This paper makes the following **key contributions**. **a)** On the formulation side, we generalize existing models which assume an identical transformation applied to both the source/target domains to compensate for the domain shift. Our proposal permits domain-specific transformations to align both the marginal (and the conditional) data distributions; **b)** On the statistical side, we derive a provably consistent hypothesis test to check whether the transformation model can indeed correct the 'shift', directly yielding $p$-values. We also show consistency of the model in that we can provably estimate the actual transformation parameters in an asymptotic sense; **c)** We identify some interesting links of our estimation with recent developments in continuous optimization and show how our model permits an analysis based on obtaining successively tighter lower bounds; **d)** Finally, we present experiments on an AD study showing how CSF data from different batches (source/target) can be harmonized enabling the application of standard statistical analysis schemes.

## 2 Background

Consider the unsupervised domain adaptation setting where the inputs/features/covariates in the source and target domains are denoted by $\mathbf{x}_s$ and $\mathbf{x}_t$ respectively. The source and target feature spaces are related via some unknown mapping, which is recovered by applying some appropriate transformations on the inputs. We denote these transformed inputs as $\tilde{\mathbf{x}}_s$ and $\tilde{\mathbf{x}}_t$. Within this setting, our goal is two-fold: first, to *estimate* the source-to-target mapping, followed by performing some statistical *test* about the 'goodness' of the estimate. Specifically, the problem is to first estimate suitable transformations $h \in \mathcal{G}$, $g \in \mathcal{G}'$, parameterized by some $\lambda$ and $\beta$ respectively, such that the transformed data $\tilde{\mathbf{x}}_s := h(\mathbf{x}_s, \lambda)$ and $\tilde{\mathbf{x}}_t := g(\mathbf{x}_t, \beta)$ have *similar* distributions. $\mathcal{G}$ and $\mathcal{G}'$ restrict the allowable mappings (e.g., affine) between source and target. Clearly the goodness of domain adaptation depends on the nature and size of $\mathcal{G}$, and the similarity measure used to compare the distributions. The distance/similarity measure used in our model defines a statistic for comparing distributions. Hence, using the estimated transformations, we then provide a hypothesis test for the existence of $\lambda$ and $\beta$ such that $Pr(\tilde{\mathbf{x}}_s) = Pr(\tilde{\mathbf{x}}_t)$, and finally assign $p$-values for the significance.

To setup this framework, we start with a statistic that measures the distance between two distributions. As motivated in Section 1, we do not impose any parametric assumptions. Since we are interested in the mismatch of $Pr(\tilde{\mathbf{x}}_s)$ and $Pr(\tilde{\mathbf{x}}_t)$, we use maximum mean discrepancy (MMD) which measures the mean distance between $\{\mathbf{x}_s\}$ and $\{\mathbf{x}_t\}$ in a Hilbert space induced by a characteristic kernel $\mathcal{K}$,

$$MMD(\mathbf{x}_s, \mathbf{x}_t) = \sup_{f \in F} \left( \frac{1}{m} \sum_{i=1}^{m} f(\mathbf{x}_s^i) - \frac{1}{n} \sum_{i=1}^{n} f(\mathbf{x}_t^i) \right) = \| \frac{1}{m} \sum_{i=1}^{m} \mathcal{K}(\mathbf{x}_t^i, \cdot) - \frac{1}{n} \sum_{i=1}^{n} \mathcal{K}(\mathbf{x}_s^i, \cdot) \|_H \quad (1)$$

where $F = \{f \in H_{\mathcal{K}}, ||f||_{H_{\mathcal{K}}} \leq 1\}$ and $H_{\mathcal{K}}$ denotes the universal RKHS. The advantage of MMD over other nonparametric distance measures is discussed in [30, 15, 16, 31]. Specifically, MMD statistic defines a metric, and whenever MMD is large, the samples are "likely" from different distributions. The simplicity of MMD and the statistical and asymptotic guarantees it provides [15, 16], largely drive our estimation and testing approach. In fact, our framework will operate on 'transformed' data $\tilde{\mathbf{x}}_s$ and $\tilde{\mathbf{x}}_t$ while estimating the appropriate transformations.

## 2.1 Related Work

The body of work on domain adaptation is fairly extensive, even when restricted to the unsupervised version. Below, we describe algorithms that are more closely related to our work and identify the similarities/differences. A common feature of many unsupervised methods is to match the feature/covariate distributions between the source and the target domains, and broadly, these fall into two different categories. The first set of methods deal with feature distributions that may be different but not due to the distortion of the inputs/features. Denoting the labels/outputs for the source and target domains as $\mathbf{y}_s$ and $\mathbf{y}_t$ respectively, here we have, $Pr(\mathbf{y}_s|\mathbf{x}_s) \approx Pr(\mathbf{y}_t|\mathbf{x}_t)$ but $Pr(\mathbf{x}_s) \neq Pr(\mathbf{x}_t)$ – this is sampling bias. The ideas in [19, 25, 2, 5] address this by 're-weighting' the source instances so as to minimize feature distribution differences between the source and the target. Such re-weighting schemes do not necessarily correspond to transforming the source and target inputs, and may simply scale or shift the appropriate loss functions. The central difference among these approaches is the distance metric used to measure the discrepancy of the feature distributions.

The second set of methods correspond to the case where distributional differences are mainly caused by feature distortion such as change in pose, lighting, blur and resolution in visual recognition. Under this scenario, $Pr(\mathbf{y}_s|\mathbf{x}_s) \neq Pr(\mathbf{y}_t|\mathbf{x}_t)$ but $Pr(\tilde{\mathbf{x}}_s) \approx Pr(\tilde{\mathbf{x}}_t)$ and the transformed conditional distributions are close. [26, 1, 10, 14, 12] address this problem by learning the same feature transformation on source and target domains to minimize the difference of $Pr(\tilde{\mathbf{x}}_s)$ and $Pr(\tilde{\mathbf{x}}_t)$ directly. Our proposed model fits better under this umbrella — where the distributional differences are mainly caused by feature distortion due to site specific acquisition and other experimental issues. While some methods are purely data-driven such as those using geodesic flow [14, 12], backpropagation [10]) and so on, other approaches estimate the transformation that minimizes distance metrics such as the Maximum Mean Discrepancy (MMD) [26, 1]. To our knowledge, no statistical consistency results are known for any of the methods that fall in the second set.

*Overview:* The idea in [1] is perhaps the most closely related to our proposal, but with a few important differences. First, we relax the condition that the same transformation must be applied to each domain; instead, we permit domain-specific transformations. Second, we derive a provably consistent hypothesis test to check whether the transformation model can indeed correct the shift. We then prove that the model is consistent when it is correct. These theoretical results apply directly to [1], which turns out to be a special case of our framework. We find that the extension of our results to [26] is problematic since that method violates the requirement that the mean differences should be measured in a valid Reproducing Kernel Hilbert space (RKHS).

## 3 Model

We first present the objective function of our estimation problem and provide a simple algorithm to compute the unknown parameters $\lambda$ and $\beta$. Recall the definition of MMD from (1). Given the kernel $\mathcal{K}$ and the source and target inputs $\mathbf{x}_s$ and $\mathbf{x}_t$, we are interested in the MMD between the "transformed" inputs $\tilde{\mathbf{x}}_s$ and $\tilde{\mathbf{x}}_t$. We are only provided the class of the transformations; $m$ and $n$ denote the sample sizes of source and target inputs. So our objective function is simply

$$\min_{\lambda \in \Omega_\lambda} \min_{\beta \in \Omega_\beta} ||\mathbb{E}_{\mathbf{x}_t} \mathcal{K}(g(\mathbf{x}_t, \beta), \cdot) - \mathbb{E}_{\mathbf{x}_s} \mathcal{K}(h(\mathbf{x}_s, \lambda), \cdot)||_H \tag{2}$$

where $\lambda \in \Omega_\lambda$ and $\beta \in \Omega_\beta$ are the constraint sets of the unknowns. Assume that the parameters are bounded is reasonable (discussed in Section 4.3), and their approximations can be easily computed using certain data statistics. The empirical estimate of the above objective would simply be

$$\min_{\lambda \in \Omega_\lambda} \min_{\beta \in \Omega_\beta} ||\frac{1}{m} \sum_{i=1}^{m} \mathcal{K}(g(\mathbf{x}_t^i, \beta), \cdot) - \frac{1}{n} \sum_{i=1}^{n} \mathcal{K}(h(\mathbf{x}_s^i, \lambda), \cdot)||_H \tag{3}$$

*Remarks:* We note a few important observations about (3) to draw the contrast from (1). The power of MMD lies in differentiating feature distributions, and the correction factor is entirely dependant on the choice of the kernel class – a richer one does a better job. Instead, our objective in (3) is showing that complex distortions *can* be corrected *before* applying the kernel in an intra-domain manner (as we show in Section 4). From the perspective of the complexity of distortions, this strategy may correspond to a larger hypotheses space compared to the classical MMD setup. This is clearly beneficial in settings where source and target are related by complex feature distortions.

It may be seen from the structure of the objective in (3) that designing an algorithm for any given $\mathcal{K}$ and $\mathcal{G}$ may not be straightforward. We present the estimation procedure for certain widely-used classes of $\mathcal{K}$ and $\mathcal{G}$ in Section 4.3. For the remainder of the section, where we present our testing procedure and describe technical results, we will assume that we can solve the above objective and the corresponding estimates are denoted by $\hat{\lambda}$ and $\hat{\beta}$.

### 3.1 Minimal MMD test statistic

Observe that the objective in (3) is based on the assumption that the transformations $h(\cdot) \in \mathcal{G}$ and $g(\cdot) \in \mathcal{G}'$ ($\mathcal{G}$ and $\mathcal{G}'$ may be different if desired) are sufficient in some sense for 'correcting' the discrepancy between the source and target inputs. Hence, we need to specify a model checking task on the recoverability of these transforms, while also concurrently checking the goodness of the estimates of $\lambda$ and $\beta$. This task will correspond to a hypothesis test where the two hypotheses being compared are as follows.

**H₀** : There exists a $\lambda$ and $\beta$ such that $Pr(g(\mathbf{x}_t, \beta)) = Pr(h(\mathbf{x}_s, \lambda))$.

**H_A** : There does not exist any such $\lambda$ and $\beta$ such that $Pr(g(\mathbf{x}_t, \beta)) = Pr(h(\mathbf{x}_s, \lambda))$.

Since the statistic for testing **H₀** here needs to measure the discrepancy of $Pr(g(\mathbf{x}_t, \beta))$ and $Pr(h(\mathbf{x}_s, \lambda))$, one can simply use the objective from (3). Hence our test statistic is given by the *minimal* MMD estimate for a given $h \in \mathcal{G}$, $g \in \mathcal{G}'$, $\mathbf{x}_s$, $\mathbf{x}_t$ and computed at the estimates $\hat{\lambda}$, $\hat{\beta}$

$$(\hat{\lambda}, \hat{\beta}) := \arg \min_{\lambda \in \Omega_\lambda} \min_{\beta \in \Omega_\beta} \mathcal{M}(\lambda, \beta) := \| \frac{1}{m} \sum_{i=1}^{m} \mathcal{K}(g(\mathbf{x}_t^i, \beta), \cdot) - \frac{1}{n} \sum_{i=1}^{n} \mathcal{K}(h(\mathbf{x}_s^i, \lambda), \cdot) \|_H \qquad (4)$$

We denote the population estimates of the parameters under the null and alternate hypothesis as $(\lambda_0, \beta_0)$ and $(\lambda_A, \beta_A)$. Recall that the MMD corresponds to a statistic, and it has been used for testing the equality of distributions in earlier works [15]. It is straightforward to see that the *true* minimal MMD $\mathcal{M}^*(\lambda_0, \beta_0) = 0$ if and only if **H₀** is true. Observe that (4) is the empirical (and hence biased) 'approximation' of the true minimal MMD statistic $\mathcal{M}^*(\cdot)$ from the objective in (2). This will be used while presenting our technical results (in Section 4) on the consistency and the corresponding statistical power guaranteed by this *minimal MMD statistic* based testing.

**Relationship to existing approaches.** Hypothesis testing involves transforming the inputs before comparing their distributions in some RKHS (while we solve for the transformation parameters). The approach in [15, 16] applies the kernel to the input data directly and asks whether or not the distributions are the same based on the MMD measure. Our approach derives from the intuition that allowing for the two-step procedure of transforming the inputs first, followed by computing their distance in some RKHS is flexible, and in some sense is more general compared to directly using MMD (or other distance measures) on the inputs. To see this, consider the simple example where $\mathbf{x}_s \sim \mathcal{N}(0, 1)$ and $\mathbf{x}_t = \mathbf{x}_s + 1$. A simple application of MMD (from (1)) on the inputs $\mathbf{x}_s$ and $\mathbf{x}_t$ directly will reject the null hypothesis (where the $H_0$ states that the source and target are the same distributions). Our algorithm allows for a transformation on the source/target and will correct this discrepancy and accept **H₀**. Further, our proposed model generalizes the approach taken in [1]. Specifically, their approach is a special case of (3) with $h(\mathbf{x}_s) = \mathbf{W}^T \mathbf{x}_s$, $g(\mathbf{x}_t) = \mathbf{W}^T \mathbf{x}_t$ ($\lambda$ and $\beta$ correspond to $\mathbf{W}$ here) with the constraint that $\mathbf{W}$ is orthogonal.

*Summary:* Overall, our estimation followed by testing procedure will be two-fold. Given $\mathbf{x}_s$ and $\mathbf{x}_t$, the kernel $\mathcal{K}$ and the function spaces $\mathcal{G}, \mathcal{G}'$, we first estimate the unknowns $\lambda$ and $\beta$ (described in Section 4.3). The corresponding statistic $\mathcal{M}(\hat{\lambda}, \hat{\beta})$ at the estimates is then compared to a given significance threshold $\gamma$. Whenever $\mathcal{M}(\hat{\lambda}, \hat{\beta}) > \gamma$ the null **H₀** is rejected. This rejection simply indicates that $\mathcal{G}$ and/or $\mathcal{G}'$ are not sufficient in recovering the mismatch of source to target at

the Type I error of $\alpha$. Clearly, the richness of these function classes is central to the power of the testing procedure. We will further argue in Section 4 that even allowing $h(\cdot)$ and $g(\cdot)$ to be linear transformations greatly enhances the ability to remove the distorted feature distributions and reliably test their difference or equivalence. Also the test is non-parametric and handles missing (systematic/noisy) features among the two distributions of interest (see appendix for more details).

## 4 Consistency

Building upon the two-fold estimating and testing procedure presented in the previous sections, we provide several guarantees about the estimation consistency and the power of minimal MMD based hypothesis testing, both in the asymptotic and finite sample regimes. The technical results presented here are applicable for large classes of transformation functions $\mathcal{G}$ with fairly weak and reasonable assumptions on $\mathcal{K}$. Specifically we consider Holder-continuous $h(\cdot)$ and $g(\cdot)$ functions on compact sets $\Omega_\lambda$ and $\Omega_\beta$. Like [15], we have $\mathcal{K}$ to be a bounded non-negative characteristic kernel i.e., $0 \leq \mathcal{K}(\mathbf{x}, \mathbf{x}') \leq K \ \forall \mathbf{x}, \mathbf{x}'$, and we assume $\partial \mathcal{K}$ to be bounded in a neighborhood of 0. We note that technical results for an even more general class of kernels are fairly involved and so in this paper we restrict ourselves to radial basis kernels. Nevertheless, even under the above assumptions our null hypothesis space is more general than the one considered in [15] because of the extra transformations that we allow on the inputs. With these assumptions, and the Holder-continuity of $h(\mathbf{x}_s, \cdot)$ and $g(\mathbf{x}_t, \cdot)$, we assume

**(A1)** $\quad \|\mathcal{K}(h(\mathbf{x}_s, \lambda_1), \cdot) - \mathcal{K}(h(\mathbf{x}_s, \lambda_2), \cdot)\| \leq L_h d(\lambda_1, \lambda_2)^{r_h} \quad \forall \mathbf{x}_s; \lambda_1, \lambda_2 \in \Omega_\lambda$

**(A2)** $\quad \|\mathcal{K}(g(\mathbf{x}_t, \beta_1), \cdot) - \mathcal{K}(g(\mathbf{x}_t, \beta_2), \cdot)\| \leq L_g d(\beta_1, \beta_2)^{r_g} \quad \forall \mathbf{x}_t; \beta_1, \beta_2 \in \Omega_\beta$

### 4.1 Estimation Consistency

Observe that the minimization of (3) assumes that the null is true i.e., the parameter estimates correspond to $\mathbf{H_0}$. Therefore, we discuss consistency in the context of existence of a unique set of parameters $(\lambda_0, \beta_0)$ that match the distributions of $\tilde{\mathbf{x}}_s$ and $\tilde{\mathbf{x}}_t$ perfectly. By inspecting the structure of the objective in (2) and (3), we see that the estimates will be asymptotically unbiased. Our first set of results summarized here provide consistency of the estimation whenever the assumptions **(A1)** and **(A2)** hold. This consistency result follows from the convergence of objective. All the proofs are included in the appendix.

**Theorem 4.1** (**MMD Convergence**). *Under* $\mathbf{H_0}$, $\|\mathbb{E}_{x_s}\mathcal{K}(h(\mathbf{x}_s, \hat{\lambda}), \cdot) - \mathbb{E}_{x_t}\mathcal{K}(g(\mathbf{x}_t, \hat{\beta}), \cdot)\|_H \to 0$ *at the rate,* $\max\left(\frac{\sqrt{\log n}}{\sqrt{n}}, \frac{\sqrt{\log m}}{\sqrt{m}}\right)$.

**Theorem 4.2** (**Consistency**). *Under* $\mathbf{H_0}$, *the estimators* $\hat{\lambda}$ *and* $\hat{\beta}$ *are consistent.*

*Remarks:* Theorem 4.1 shows the convergence rate of MMD distance between the source and the target after the transformations are applied. Recall that $m$ and $n$ are the sample sizes of source and target respectively, and $h(\mathbf{x}_s, \hat{\lambda})$ and $g(\mathbf{x}_t, \hat{\beta})$ are the recovered transformations.

### 4.2 Power of the Hypothesis Test

We now discuss the statistical power of minimal MMD based testing. The next set of results establish that the testing setup from Section 3.1 is asymptotically consistent. Recall that $\mathcal{M}^*(\cdot)$ denotes the (unknown) expected statistic from (2) while $\mathcal{M}(\cdot)$ is its empirical estimate from (4).

**Theorem 4.3** (**Hypothesis Testing**). *(a) Whenever* $\mathbf{H_0}$ *is true, with probability at least* $1 - \alpha$,

$$0 \leq \mathcal{M}(\hat{\lambda}, \hat{\beta}) \leq \sqrt{\frac{2K(m+n)\log\alpha^{-1}}{mn}} + \frac{2\sqrt{K}}{\sqrt{n}} + \frac{2\sqrt{K}}{\sqrt{m}} \tag{5}$$

*(b) Whenever* $\mathbf{H_A}$ *is true, with probability at least* $1 - \epsilon$,

$$\mathcal{M}(\hat{\lambda}, \hat{\beta}) \leq \mathcal{M}^*(\lambda_A, \beta_A) + \sqrt{\frac{2K(m+n)\log\epsilon^{-1}}{mn}} + \frac{2\sqrt{K}}{\sqrt{n}} + \frac{2\sqrt{K}}{\sqrt{m}}$$

$$\mathcal{M}(\hat{\lambda}, \hat{\beta}) \geq \mathcal{M}^*(\lambda_A, \beta_A) - \frac{\sqrt{K}}{\sqrt{n}}\left(4 + \sqrt{C^{(h,\epsilon)} + \frac{d_\lambda}{2r_h}\log n}\right) - \frac{\sqrt{K}}{\sqrt{m}}\left(4 + \sqrt{C^{(g,\epsilon)} + \frac{d_\beta}{2r_g}\log m}\right)$$

$$\tag{6}$$

*where $C^{(h,\epsilon)} = \log(2|\Omega_\lambda|) + \log \epsilon^{-1} + \frac{d_\lambda}{r_h} \log \frac{L_h}{\sqrt{K}}$, and $C^{(g,\epsilon)} = \log(2|\Omega_\beta|) + \log \epsilon^{-1} + \frac{d_\beta}{r_g} \log \frac{L_g}{\sqrt{K}}$*

*Remarks:* We make a few comments about the theorem. Recall that the constant $K$ is the kernel bound, and $L_h$, $L_g$, $r_h$ and $r_g$ are defined in (A1)(A2). $d_\lambda$ and $d_\beta$ are the dimensions of $\lambda$ and $\beta$ spaces respectively. Observe that whenever $\mathbf{H_0}$ is true, (5) shows that $\mathcal{M}(\hat{\lambda}, \hat{\beta})$ approaches 0 as the sample size increases. Similarly, under $\mathbf{H_A}$ the statistic converges to some positive (unknown) value $\mathcal{M}^*(\lambda_A, \beta_A)$. Following these observations, Theorem 4.3 basically implies that the statistical power of our test (described in Section 3.1) increases to 1 as the sample size $m, n$ increases. Except constants, the upper bounds under both $H_0$ and $H_A$ have a rate of $\max(\frac{1}{\sqrt{n}}, \frac{1}{\sqrt{m}})$, while the lower bound under $H_A$ has the rate $\max(\frac{\sqrt{\log n}}{\sqrt{n}}, \frac{\sqrt{\log m}}{\sqrt{m}})$. In the appendix we show that (see Lemma 4.5) as $m, n \to \infty$, the constants $|\Omega_\lambda|$, $|\Omega_\beta|$ converge to a small positive number, thus removing the dependence of consistency on these constants.

The dependence on the sizes of search spaces $\Omega_\lambda$ and $\Omega_\beta$ may nevertheless make the bounds for $\mathbf{H_A}$ loose. In practice, one can choose 'good' bound constraints based on some pre-processing on the source and target inputs (e.g., comparison of median and modes). The loss in power due to overestimated $\Omega_\lambda$ and $\Omega_\beta$ will be compensated by 'large enough' sample sizes. Observe that this trade-off of sample size versus complexity of hypothesis space is fundamental in statistical testing and is not specific to our model. We further investigate this trade-off for certain special cases of transformations $h(\cdot)$ and $g(\cdot)$ that may be of interest in practice. For instance, consider the scenario where one of the transformations is identity and the other one is linear in the unknowns. Specifically, $\tilde{\mathbf{x}}_t = \mathbf{x}_t$ and $h_0(\mathbf{x}_s, \lambda_0) = \phi(\mathbf{x}_s)^T \lambda_0$ where $\phi(\cdot)$ is some known transformation. Although restrictive, this scenario is very common in medical data acquisition (refer to Section 1) where the source and target inputs are assumed to have linear/affine distortions. Within this setting, the assumptions for our technical results will be satisfied whenever $\phi(\mathbf{x}_s)$ is bounded with high probability and with $r_h = \frac{1}{2}$. We have the following result for this scenario (Var($\cdot$) denotes empirical variance).

**Theorem 4.4 (Linear transformation).** *Under $\mathbf{H_0}$, identity $g(\cdot)$ with $h = \phi(\mathbf{x}_s)^T \lambda$, we have $\Omega_\lambda := \{\lambda; |\frac{1}{n} \sum_{i=1}^{n} \|\mathbf{x}_t^i - \phi(\mathbf{x}_s^i)^T \lambda)\|^2 \le 3 \sum_{k=1}^{p} Var(\mathbf{x}_{t,k}) + \epsilon\}$. For any $\epsilon, \alpha > 0$ and sufficiently large sample size, a neighborhood of $\lambda_0$ is contained in $\Omega_\lambda$ with probability at least $1 - \alpha$.*

Observe that subscript $k$ in $\mathbf{x}_{t,k}$ above denotes the $k^{th}$ dimensional feature of $\mathbf{x}_t$. The above result implies that the search space for $\lambda$ reduces to a quadratic constraint in the above described example scenario. Clearly, this refined search region would enhance the statistical power for the test even when the sample sizes are small (which is almost always the case in population studies). Note that such refined sets may be computed using 'extra' information about the structure of the transformations and/or input data statistics, there by allowing for better estimation and higher power. Lastly, we point out that the ideas presented in [16] for a finite sample testing setting translate to our model as well but we do not present explicit details in this work.

### 4.3 Optimization Lower Bounds

We see that it is valid to assume that the feasible set is compact and convex for our purposes (Theorem 4.4). This immediately allows us to use algorithms that exploit feasible set compactness to estimate model parameters, for instance, conditional gradient algorithms which have low per iteration complexity [23]. Even though these algorithms offer practical benefits, with non-convex objective, it is nontrivial to analyze their theoretical/convergence aspects, and as was noted earlier in Section 3, except for simplistic $\mathcal{G}$, $\mathcal{G}'$ and $\mathcal{K}$, the minimization in (3) might involve a non-convex objective. We turn to some recent results which have shown that *specific classes of non-convex problems* or NP-Hard problems can be solved to any desired accuracy using a sequence of convex optimization problems [33]. This strategy is currently an active area of research and has already shown to provide impressive performance in practice [3].

Very recently,[4] showed that one such class of problems called *signomial* programming can be solved using successive relative entropy relaxations. Interestingly, we show that for the widely-used class of Gaussian kernels, our objective can be optimized using these ideas. For notational simplicity, we do not transform the targets i.e, $\tilde{\mathbf{x}}_t = \mathbf{x}_t$ or $g(\cdot)$ is identity and only allow for linear transformations $h(\cdot)$. Observe that, with respect to the estimation problem (refer to (3)) this is the same as transforming both source and target inputs. When $\mathcal{K}$ is Gaussian, the objective in (3) with identity $g(\cdot)$ and linear

$h(\cdot)$ ($\lambda$ corresponds to slope and intercept here) can be equivalently written as,

$$\min_{\lambda \in \Omega_\lambda} \left( \frac{1}{n^2} \sum_{i=1}^{n} \sum_{j=1}^{n} \mathcal{K}(h(\mathbf{x}_s^i, \lambda), h(\mathbf{x}_s^j, \lambda)) - \frac{2}{mn} \sum_{i=1}^{m} \sum_{j=1}^{n} \mathcal{K}(\mathbf{x}_t^i, h(\mathbf{x}_s^j, \lambda)) \right)$$
$$:= \min_{\lambda \in \Omega_\lambda} \sum_j \frac{1}{n^2} \exp\left( - \left( a_j^T \lambda \lambda^T a_i \right) \right) - \sum_{i,j} \frac{2}{mn} \exp\left( - \left( b_{ij}^T \lambda \lambda^T b_{ij} + 2cb_{ij}^T \lambda + c^2 \right) \right) \tag{7}$$

for appropriate $a_j$, $b_{ij}$ and $c$. Denoting $\gamma = \lambda \lambda^T$, the above objective can be made linear in the decision variables $\gamma$ and $\lambda$ thus putting it in the standard form of signomial optimization. The convex relaxation of the quadratic equality constraint is $\gamma - \lambda \lambda^T \succeq 0$, hence we seek to solve,

$$\min_{\gamma, \lambda} \sum_j \frac{1}{n^2} \exp\left( \mathrm{tr}(A_j \gamma) \right) - \sum_{i,j} \frac{2}{mn} \exp\left( \mathrm{tr}(B_{ij}\gamma) + C_{ij}^T \lambda + c \right) \quad \text{s.t.} \quad \gamma - \lambda \lambda^T \succeq 0 \tag{8}$$

Clearly the objective is exactly in the form that [4] solves, albeit we also have a convex constraint. Nevertheless, using their procedure for the unconstrained signomial optimization we can write a sequence of convex relaxations for this objective. This sequence is hierarchical, in the sense that, as we go down the sequence, each problem gives tighter bounds to the original nonconvex objective [4]. For our applications, we see that since confidence interval procedure (mentioned earlier) naturally suggests a good initial point in addition, any generic (local) numerical optimization schemes like trust region, gradient projection etc. can be used to solve (7) whereas the hierarchy of (8) can be used in general when one does not have access to a good starting point.

## 5 Experiments

**Design and Overall Goals.** We performed evaluations on both synthetic data as well as data from an AD study. **(A)** We first evaluate the goodness of our estimation procedure and the power of the minimal MMD based test when the source and target inputs are *known* transformations of samples from different distribution families (e.g., Normal, Laplace). Here, we seek to clearly identify the influence of the sample size as well as the effect of the transformations on recoverability. **(B)** After these checks, we then apply our proposed model for matching CSF protein levels of 600 subjects. These biomarkers were collected in two different batches; it is known that the measures for the same participant (across batches) have high variability [20]. In our data, fortunately, a subset of individuals have *both* batch data (the "real" measurement must be similar in both batches) whereas a fraction of individuals' CSF is only available in one batch. If we find a linear standardization between the

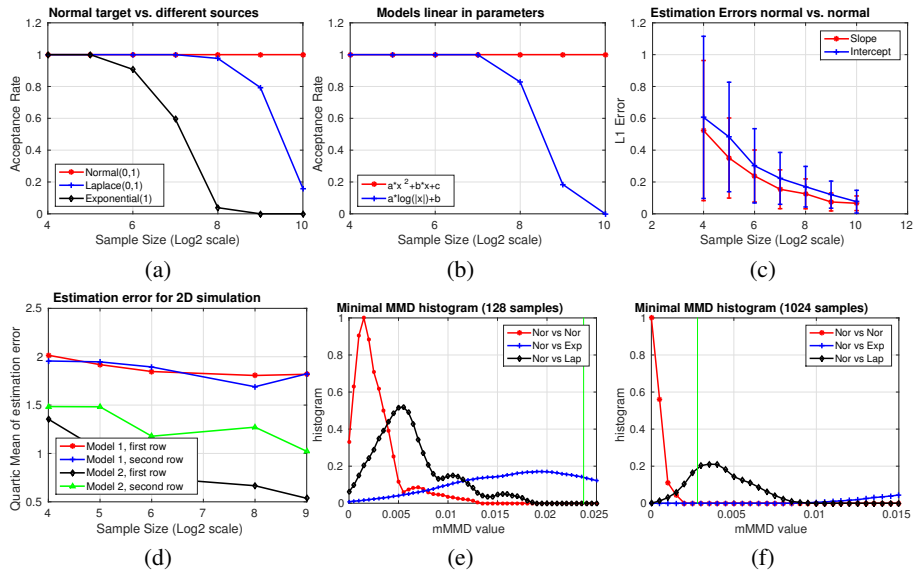

Figure 1: (a,b) Acceptance Ratios, (c,d) Estimation errors, (e,f) Histograms of minimal MMD statistic.

two batches it serves as a gold standard, against which we compare our algorithm which does *not* use information about corresponding samples. Note that the standardization trick is unavailable in multi-center studies; we use this data in this paper simply to make the description of our evaluation design simpler which, for multi-site data pooling, must be addressed using secondary analyses.

**Synthetic data.** Fig 1 summarizes our results on synthetic data where the source are Normal samples and targets comes from different families. First, observe that our testing procedure efficiently rejects $H_0$ whenever the targets are not Normal (blue and black curves in Fig. 1(a)). If the transformation class is beyond linear (e.g., log), the null is efficiently rejected as samples increase (see Fig. 1(b)). Beyond the testing power, Figs. 1(c,d) shows the error in the actual estimates, which decrease as the sample size increases (with tighter confidence intervals). The appendix includes additional model details. To get a better idea about the minimal MMD statistic, we show its histogram (over multiple bootstrap simulations) for different targets in Fig 1(e,f). The green line here denotes the bootstrap significance threshold (0.05). In Fig. 1(e,f), the red curve is always to the left of the threshold, as desired. However, the samples are not enough to reject the null the black and blue curves; and we will need larger

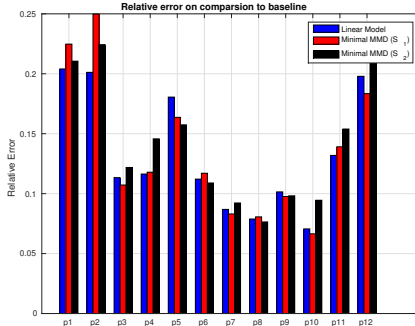

Figure 2: Relative error in transformation estimation between CSF batches.

sample sizes (Fig. 1(f)). If needed, the minimal MMD value can be used to obtain a better threshold. Overall, these plots show that the minimal MMD based estimation and testing setup robustly removes the feature distortions and facilitates the statistical test.

**AD study.** Fig 2 shows the relative errors after correcting the feature distortions between the two batches in the 12 CSF proteins. The bars correspond to simple linear "standardization" transformation where we assume we have corresponding sample information (blue) and our minimal MMD based domain adaptation procedure on sets $S_1$ and $S_2$ ($S_1$: participants available in both batches, $S_2$: all participants). Our models perform as well as the gold standard (where some subjects have *volunteered* CSF sampling for both batches). Specifically, the trends in Fig 2 indicate that our minimal MMD based testing procedure is a powerful procedure for conducting analyses on such pooled datasets.

To further validate these observations, we used the 'transformed' CSF data from the two batches (our algorithm and gold standard) and performed a multiple regression to predict Left and Right Hippocampal Volume (which are known to be AD markers). Table 1 shows that the correlations (predicted vs. actual) resulting from the minimal MMD corrected data are comparable or offer improvements to the alternatives. We point out that the best correlations are achieved when all the data is used with minimal MMD (which the gold standard cannot benefit from). Any downstream prediction tasks we wish to conduct are independent of the model presented here.

Table 1: Performance of transformed (our vs. gold standard) CSF on a regression task.

| Model | Left | Right |
|---|---|---|
| None | 0.46± 0.15 | 0.37±0.16 |
| Linear | 0.46± 0.15 | 0.37±0.16 |
| $\mathcal{M}(S_1)$ | **0.48± 0.15** | **0.39± 0.15** |
| $\mathcal{M}(S_2)$ | **0.48± 0.15** | **0.40± 0.15** |

## 6  Conclusions

We presented a framework for kernelized statistical testing on data from multiple sources when the observed measurements/features have been systematically distorted/transformed. While there is a rich body of work on kernel test statistics based on the maximum mean discrepancy and other measures, the flexibility to account for a given class of transformations offers improvements in statistical power. We analyze the statistical properties of the estimation and demonstrate how such a formulation may enable pooling datasets from multiple participating sites, and facilitate the conduct of neuroscience studies with substantially higher sample sizes which may be otherwise infeasible.

**Acknowledgments:** This work is supported by NIH AG040396, NIH U54AI117924, NSF DMS1308847, NSF CAREER 1252725, NSF CCF 1320755 and UW CPCP AI117924. The authors are grateful for partial support from UW ADRC AG033514 and UW ICTR 1UL1RR025011. We thank Marilyn S. Albert (Johns Hopkins) and Anne Fagan (Washington University at St. Louis) for discussions at a preclinical Alzheimer's disease meeting in 2015 (supported by Stay Sharp fund).

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
