[Supplementary Material]

# Hypothesis Testing in Unsupervised Domain Adaptation with Applications in Alzheimer's Disease (Supplementary Material)

**Hao Henry Zhou**[†]   **Sathya N. Ravi**[†]   **Vamsi K. Ithapu**[†]

**Sterling C. Johnson**[§†]   **Grace Wahba**[†]   **Vikas Singh**[†]

[§]William S. Middleton Memorial VA Hospital   [†]University of Wisconsin–Madison
hzhou@stat.wisc.edu   ravi5@wisc.edu   vamsi@cs.wisc.edu
scj@medicine.wisc.edu   wahba@stat.wisc.edu   vsingh@biostat.wisc.edu

## 1   Preparatory Results

**McDiarmid's Inequality** Let $f : X^m \to R$ be a function such that for all $i \in \{1, ..., m\}$, there exists $c_i < \infty$ for which

$$\sup_{x \in X^m, \tilde{x} \in X} |f(x_1, ..., x_m) - f(x_1, ..., x_{i-1}, \tilde{x}, x_{i+1}, ..., x_m)| \le c_i \tag{1}$$

Then for all probability measures p and every $\epsilon > 0$,

$$Pr(f(x) - E_x(f(x)) > \epsilon) < \exp(-\frac{2\epsilon^2}{\sum_{i=1}^m c_i{}^2}) \tag{2}$$

where $E_x$ denotes the expectation over the $m$ random variables $x_i \sim p$, and $Pr$ denotes the probability over these m variables.

**Lemma 1.1.** *For any fixed function $h(x_s, \lambda)$, $g(x_t, \beta)$, any $\lambda$, $\beta$, bounded kernel $\mathcal{K}$, we have*

$$P(\sup_{f \in F} |\frac{1}{n} \sum_{i=1}^n f(h(x_s^i, \lambda)) - E_{x_s} f(h(x_s, \lambda))| - E_{x_s} \sup_{f \in F} |\frac{1}{n} \sum_{i=1}^n f(h(x_s^i, \lambda)) - E_{x_s} f(h(x_s, \lambda))| > \epsilon) \tag{3}$$

$$\le \exp\left(-\frac{\epsilon^2 n}{2K}\right) \tag{4}$$

$$P(\sup_{f \in F} |\frac{1}{m} \sum_{i=1}^m f(g(x_t^i, \beta)) - E_{x_t} f(g(x_t, \beta))| - E_{x_t} \sup_{f \in F} |\frac{1}{m} \sum_{i=1}^m f(g(x_t^i, \beta)) - E_{x_t} f(g(x_t, \beta))| > \epsilon) \tag{5}$$

$$\le \exp\left(-\frac{\epsilon^2 m}{2K}\right) \tag{6}$$

*Proof.* When we replace $x_s^i$ by $\tilde{x}_s^i$ and $x_s$ by $\tilde{x}_s$, we have

$$\sup_{\tilde{x}_s} |(\sup_{f \in F} |\frac{1}{n} \sum_{i=1}^{n} f(h(x_s^i, \lambda)) - E_{x_s} f(h(x_s, \lambda))|) - (\sup_{f \in F} |\frac{1}{n} \sum_{i=1}^{n} f(h(\tilde{x}_s^i, \lambda)) - E_{x_s} f(h(x_s, \lambda))|)| \tag{7}$$

We use the fact that $|\sup_{x} f_1(x) - \sup_{x} f_2(x)|$ is smaller than $\sup_{x} |f_1(x) - f_2(x)|$, and get

$$\leq \sup_{\tilde{x}_s} \sup_{f \in F} |(\frac{1}{n} \sum_{i=1}^{n} f(h(x_s^i, \lambda)) - E_{x_s} f(h(x_s, \lambda))) - (\frac{1}{n} \sum_{i=1}^{n} f(h(\tilde{x}_s^i, \lambda)) - E_{x_s} f(h(x_s, \lambda)))| \tag{8}$$

$$= \sup_{\tilde{x}_s} \sup_{f \in F} |\frac{1}{n} (f(h(x_s^i, \lambda)) - f(h(\tilde{x}_s^i, \lambda)))| \tag{9}$$

It follows from Riesz representation theorem in reproducing Hilbert space that

$$= \frac{1}{n} \sup_{\tilde{x}_s} \|\mathcal{K}(h(x_s^i, \lambda), .) - \mathcal{K}(h(\tilde{x}_s^i, \lambda), .)\|_H \tag{10}$$

It follows from kernel bounded property that

$$\leq \frac{2\sqrt{K}}{n} \tag{11}$$

Now using McDiarmid's Inequality, we have

$$P(\sup_{f \in F} |\frac{1}{n} \sum_{i=1}^{n} f(h(x_s^i, \lambda)) - E_{x_s} f(h(x_s, \lambda))| - E_{x_s} \sup_{f \in F} |\frac{1}{n} \sum_{i=1}^{n} f(h(x_s^i, \lambda)) - E_{x_s} f(h(x_s, \lambda))| > \epsilon) \tag{12}$$

$$\leq \exp(-\frac{\epsilon^2 n}{2K}) \tag{13}$$

A similar proof holds for $g(x_t, \beta)$. This proof can also be found in [1]. $\qquad \square$

**Lemma 1.2.** *For any fixed function $h(x_s, \lambda)$, $g(x_t, \beta)$, any $\lambda$, $\beta$, bounded kernel $\mathcal{K}$, we have*

$$E_{x_s} \sup_{f \in F} |\frac{1}{n} \sum_{i=1}^{n} f(h(x_s^i, \lambda)) - E_{x_s} f(h(x_s, \lambda))| \leq \frac{2\sqrt{K}}{\sqrt{n}} \tag{14}$$

$$E_{x_t} \sup_{f \in F} |\frac{1}{m} \sum_{i=1}^{m} f(g(x_t^i, \beta)) - E_{x_t} f(g(x_t, \beta))| \leq \frac{2\sqrt{K}}{\sqrt{m}} \tag{15}$$

*Proof.* Let $x_s'$ be the samples generated from the same distribution as $x_s$. Also, $\sigma_i i.i.d \sim \{-1, 1\}$ with equal probability. We can show,

$$E_{x_s} \sup_{f \in F} |\frac{1}{n} \sum_{i=1}^{n} f(h(x_s^i, \lambda)) - E_{x_s} f(h(x_s, \lambda))| \tag{16}$$

$$= E_{x_s} \sup_{f \in F} |E_{x_s'} (\frac{1}{n} \sum_{i=1}^{n} f(h(x_s'^i, \lambda)) - \frac{1}{n} \sum_{i=1}^{n} f(h(x_s^i, \lambda)))| \tag{17}$$

It follows from the convexity of the absolute function, and Jensen's Inequality that

$$\leq E_{x_s} \sup_{f \in F} E_{x_s'} |(\frac{1}{n} \sum_{i=1}^{n} f(h(x_s'^i, \lambda)) - \frac{1}{n} \sum_{i=1}^{n} f(h(x_s^i, \lambda)))| \tag{18}$$

$$\leq E_{x_s} E_{x_s'} \sup_{f \in F} |(\frac{1}{n} \sum_{i=1}^{n} f(h(x_s'^i, \lambda)) - \frac{1}{n} \sum_{i=1}^{n} f(h(x_s^i, \lambda)))| \tag{19}$$

For any fixed $\sigma$, the remainder have the same value because $x_s'$, $x_s$ are i.i.d samples, so

$$= E_\sigma E_{x_s} E_{x_s'} \sup_{f \in F} |\frac{1}{n} \sum_{i=1}^{n} \sigma_i (f(h(x_s'^i, \lambda)) - f(h(x_s^i, \lambda)))| \tag{20}$$

$$= E_{x_s} E_{x_s'} E_\sigma \sup_{f \in F} |\frac{1}{n} \sum_{i=1}^{n} \sigma_i (f(h(x_s'^i, \lambda)) - f(h(x_s^i, \lambda)))| \tag{21}$$

$$\leq E_{x_s} E_{x_s'} E_\sigma \sup_{f \in F} |\frac{1}{n} \sum_{i=1}^{n} \sigma_i (f(h(x_s'^i, \lambda))| + E_{x_s} E_{x_s'} E_\sigma \sup_{f \in F} |\sum_{i=1}^{n} \sigma_i f(h(x_s^i, \lambda)))| \tag{22}$$

$$= 2 E_{x_s} E_\sigma \sup_{f \in F} |\frac{1}{n} \sum_{i=1}^{n} \sigma_i f(h(x_s^i, \lambda))| \tag{23}$$

$$= \frac{2}{n} E_{x_s} E_\sigma \left\| \sum_{i=1}^{n} \sigma_i \mathcal{K}(h(x_s^i, \lambda), .) \right\|_H \tag{24}$$

$$= \frac{2}{n} E_{x_s} E_\sigma (\sum_{i,j=1}^{n} \sigma_i \sigma_j \mathcal{K}(h(x_s^i, \lambda), h(x_s^j, \lambda)))^{1/2} \tag{25}$$

It follows from Jensen's Inequality that

$$\leq E_{x_s} \left( \sum_{i,j=1}^{n} E_\sigma (\sigma_i \sigma_j) \mathcal{K}(h(x_s^i, \lambda), h(x_s^j, \lambda)) \right)^{1/2} \tag{26}$$

By using properties of $\sigma$,

$$= \frac{2}{n} E_{x_s} \left( \sum_{i=1}^{n} \mathcal{K}(h(x_s^i, \lambda), h(x_s^i, \lambda)) \right)^{1/2} \tag{27}$$

$$\leq \frac{2\sqrt{K}}{\sqrt{n}} \tag{28}$$

A similar proof holds for $g(x_t, \beta)$ and can also be found in [1]. $\qquad \square$

**Lemma 1.3.** *For any fixed function $h(x_s, \lambda)$, $g(x_t, \beta)$ and a bounded kernel $\mathcal{K}$, if (A1) holds, we have*

$$P(\sup_{\lambda \in \Omega_\lambda} \sup_{f \in F} |\frac{1}{n}\sum_{i=1}^{n} f(h(x_s^i, \lambda)) - E_{x_s}f(h(x_s, \lambda))| > \frac{\sqrt{K}}{\sqrt{n}}\left(4 + \sqrt{C^{(h,\alpha)} + \frac{d_\lambda}{2r_h}\log n}\right)) \leq \frac{\alpha}{2}$$
(29)

$$P(\sup_{\beta \in \Omega_\beta} \sup_{f \in F} |\frac{1}{m}\sum_{i=1}^{m} f(g(x_t^i, \beta)) - E_{x_t}f(g(x_t, \beta))| > \frac{\sqrt{K}}{\sqrt{m}}\left(4 + \sqrt{C^{(g,\alpha)} + \frac{d_\beta}{2r_g}\log m}\right)) \leq \frac{\alpha}{2}$$
(30)

*where $C^{(h,\alpha)} = \log(2|\Omega_\lambda|) + \log \alpha^{-1} + \frac{d_\lambda}{r_h}\log\frac{L_h}{\sqrt{K}}$, and $C^{(g,\alpha)} = \log(2|\Omega_\beta|) + \log \alpha^{-1} + \frac{d_\beta}{r_g}\log\frac{L_g}{\sqrt{K}}$*

*Proof.* Observe that we can choose a covering for $\Omega_\lambda$ using balls with any radius $\delta > 0$. Define $N_\delta$ to be number of those balls, and $\lambda_0^1, \lambda_0^2, ..., \lambda_0^{N_\delta}$ be the center of those balls. For any $d(\lambda_1, \lambda_2) \leq \delta$, we have

$$\sup_{f \in F} |\frac{1}{n}\sum_{i=1}^{n} f(h(x_s^i, \lambda_1)) - E_{x_s}f(h(x_s, \lambda_1))| - \sup_{f \in F} |\frac{1}{n}\sum_{i=1}^{n} f(h(x_s^i, \lambda_2)) - E_{x_s}f(h(x_s, \lambda_2))|$$
(31)

We use the fact that $|\sup_x f_1(x) - \sup_x f_2(x)|$ is smaller than $\sup_x |f_1(x) - f_2(x)|$

$$\leq \sup_{f \in F} |[\frac{1}{n}\sum_{i=1}^{n} f(h(x_s^i, \lambda_1)) - E_{x_s}f(h(x_s, \lambda_1))] - [\frac{1}{n}\sum_{i=1}^{n} f(h(x_s^i, \lambda_2)) - E_{x_s}f(h(x_s, \lambda_2))]|$$
(32)

$$\leq \sup_{f \in F} |\frac{1}{n}\sum_{i=1}^{n} f(h(x_s^i, \lambda_1)) - \frac{1}{n}\sum_{i=1}^{n} f(h(x_s^i, \lambda_2))| + \sup_{f \in F} |E_{x_s}f(h(x_s, \lambda_1)) - E_{x_s}f(h(x_s, \lambda_2))|$$
(33)

We use Riesz representation theorem in reproducing Hilbert space

$$= \left\|\frac{1}{n}\sum_{i=1}^{n} (\mathcal{K}(h(x_s^i, \lambda_1), .) - \mathcal{K}(h(x_s^i, \lambda_2), .))\right\|_H + \|E_{x_s}(\mathcal{K}(h(x_s, \lambda_1), .) - \mathcal{K}(h(x_s, \lambda_2), .))\|_H$$
(34)

We use Jensen's Inequality

$$\leq \frac{1}{n}\sum_{i=1}^{n} \left\|(\mathcal{K}(h(x_s^i, \lambda_1), .) - \mathcal{K}(h(x_s^i, \lambda_2), .))\right\|_H + E_{x_s} \left\|(\mathcal{K}(h(x_s, \lambda_1), .) - \mathcal{K}(h(x_s, \lambda_2), .))\right\|_H$$
(35)

$$\leq 2\sup_{x_s} \|\mathcal{K}(h(x_s, \lambda_1), .) - \mathcal{K}(h(x_s, \lambda_2), .)\|_H$$
(36)

We use (A1)

$$\leq 2L_h \delta^{r_h}$$
(37)

Thus,

$$\sup_{\lambda \in \Omega_\lambda} \sup_{f \in F} |\frac{1}{n} \sum_{i=1}^{n} f(h(x_s^i, \lambda)) - E_{x_s} f(h(x_s, \lambda))| \tag{38}$$

$$= \max_{k \in 1,2...,N_\delta} \sup_{d(\lambda, \lambda_0^k) \leq \delta} \sup_{f \in F} |\frac{1}{n} \sum_{i=1}^{n} f(h(x_s^i, \lambda)) - E_{x_s} f(h(x_s, \lambda))| \tag{39}$$

$$= \max_{k \in 1,2...,N_\delta} \sup_{d(\lambda, \lambda_0^k) \leq \delta} (\sup_{f \in F} |\frac{1}{n} \sum_{i=1}^{n} f(h(x_s^i, \lambda)) - E_{x_s} f(h(x_s, \lambda))| - \sup_{f \in F} |\frac{1}{n} \sum_{i=1}^{n} f(h(x_s^i, \lambda_0^k)) - E_{x_s} f(h(x_s, \lambda_0^k))|) \tag{40}$$

$$+ \max_{k \in 1,2...,N_\delta} \sup_{f \in F} |\frac{1}{n} \sum_{i=1}^{n} f(h(x_s^i, \lambda_0^k)) - E_{x_s} f(h(x_s, \lambda_0^k))| \tag{41}$$

$$\leq 2L_h \delta^{r_h} + \max_{k \in 1,2...,N_\delta} \sup_{f \in F} |\frac{1}{n} \sum_{i=1}^{n} f(h(x_s^i, \lambda_0^k)) - E_{x_s} f(h(x_s, \lambda_0^k))| \tag{42}$$

Thus,

$$P(\sup_{\lambda \in \Omega_\lambda} \sup_{f \in F} |\frac{1}{n} \sum_{i=1}^{n} f(h(x_s^i, \lambda)) - E_{x_s} f(h(x_s, \lambda))| > 2L_h \delta^{r_h} + \frac{2\sqrt{K}}{\sqrt{n}} + \epsilon) \tag{43}$$

$$\leq P(2L_h \delta^{r_h} + \max_{k \in 1,2...,N_\delta} \sup_{f \in F} |\frac{1}{n} \sum_{i=1}^{n} f(h(x_s^i, \lambda_0^k)) - E_{x_s} f(h(x_s, \lambda_0^k))| > 2L_h \delta^{r_h} + \frac{2\sqrt{K}}{\sqrt{n}} + \epsilon) \tag{44}$$

$$= P(\max_{k \in 1,2...,N_\delta} \sup_{f \in F} |\frac{1}{n} \sum_{i=1}^{n} f(h(x_s^i, \lambda_0^k)) - E_{x_s} f(h(x_s, \lambda_0^k))| > \frac{2\sqrt{K}}{\sqrt{n}} + \epsilon) \tag{45}$$

It follows from union bound that

$$\leq \sum_{k=1}^{N_\delta} P(\sup_{f \in F} |\frac{1}{n} \sum_{i=1}^{n} f(h(x_s^i, \lambda_0^k)) - E_{x_s} f(h(x_s, \lambda_0^k))| > \frac{2\sqrt{K}}{\sqrt{n}} + \epsilon) \tag{46}$$

$$\leq N_\delta \max_{k \in 1,2...,N_\delta} P(\sup_{f \in F} |\frac{1}{n} \sum_{i=1}^{n} f(h(x_s^i, \lambda_0^k)) - E_{x_s} f(h(x_s, \lambda_0^k))| > \frac{2\sqrt{K}}{\sqrt{n}} + \epsilon) \tag{47}$$

It follows from Lemma 1.1 and Lemma 1.2 that

$$\leq N_\delta \exp(-\frac{\epsilon^2 n}{2K}) \tag{48}$$

$$= \frac{|\Omega|}{\delta^{d_\lambda}} \exp(-\frac{\epsilon^2 n}{2K}) \tag{49}$$

where $d_\lambda$ is the dimension of $\lambda$. Here, we focus on the Euclidean space setting, but the result can directly generalized to other spaces as long as a finite covering exists for a distance measure satisfying (A1).

The results follows by setting

$$\epsilon = \sqrt{\frac{2K}{n}} \sqrt{\log(2|\Omega|) + \log \alpha^{-1} + d_\lambda \log \delta^{-1}} \tag{50}$$

$$\delta = \left(\frac{K}{L_h^2 n}\right)^{\frac{1}{2r_h}} \tag{51}$$

A similar proof holds for $g(x_t, \beta)$. □

# 4 Consistency

**Theorem 4.1 (MMD Convergence).** *Under the null hypothesis* $\mathbf{H_0}$, $\|\|\mathbb{E}_{x_s}\mathcal{K}(h(x_s, \hat{\lambda}), \cdot) - \mathbb{E}_{x_t}\mathcal{K}(g(x_t, \hat{\beta}), \cdot)\|_H - \|\mathbb{E}_{x_s}\mathcal{K}(h(x_s, \lambda_0), \cdot) - \mathbb{E}_{x_t}\mathcal{K}(g(x_t, \beta_0), \cdot)\|_H\| \to 0$ *with the rate* $\max\left(\frac{\sqrt{\log n}}{\sqrt{n}}, \frac{\sqrt{\log m}}{\sqrt{m}}\right)$.

*Proof.* Recall the basic inequality

$$\sup_{f \in F}(\frac{1}{m}\sum_{i=1}^{m} f(g(x_t^i, \hat{\beta})) - \frac{1}{n}\sum_{i=1}^{n} f(h(x_s^i, \hat{\lambda}))) \leq \sup_{f \in F}(\frac{1}{m}\sum_{i=1}^{m} f(g(x_t^i, \beta_0)) - \frac{1}{n}\sum_{i=1}^{n} f(h(x_t^i, \lambda_0))) \tag{52}$$

$$\|E_{x_s^i}\mathcal{K}(h(x_s^i, \hat{\lambda}), .) - E_{x_t}\mathcal{K}(g(x_t, \hat{\beta})\|_H - \|E_{x_s^i}\mathcal{K}(h(x_s^i, \lambda_0), .) - E_{x_t}\mathcal{K}(g(x_t, \beta_0)\|_H \tag{53}$$

$$= \sup_{f \in F}(E_{x_t}f(g(x_t, \hat{\beta})) - E_{x_s^i}f(h(x_s^i, \hat{\lambda}))) - \sup_{f \in F}(E_{x_t}f(g(x_t, \beta_0)) - E_{x_s^i}f(h(x_s^i, \lambda_0))) \tag{54}$$

We use a basic inequality and get

$$\leq \sup_{f \in F}(E_{x_t}f(g(x_t, \hat{\beta})) - E_{x_s^i}f(h(x_s^i, \hat{\lambda}))) - \sup_{f \in F}(E_{x_t}f(g(x_t, \beta_0)) - E_{x_s^i}f(h(x_s^i, \lambda_0))) \tag{55}$$

$$+ \sup_{f \in F}\frac{1}{m}\sum_{i=1}^{m} f(g(x_t^i, \beta_0)) - \frac{1}{n}\sum_{i=1}^{n} f(h(x_t^i, \lambda_0))) - \sup_{f \in F}(\frac{1}{m}\sum_{i=1}^{m} f(g(x_t^i, \hat{\beta})) - \frac{1}{n}\sum_{i=1}^{n} f(h(x_s^i, \hat{\lambda}))) \tag{56}$$

$$\leq |\sup_{f \in F}(\frac{1}{m}\sum_{i=1}^{m} f(g(x_t^i, \hat{\beta})) - \frac{1}{n}\sum_{i=1}^{n} f(h(x_s^i, \hat{\lambda}))) - \sup_{f \in F}(E_{x_t}f(g(x_t, \hat{\beta})) - E_{x_s^i}f(h(x_s^i, \hat{\lambda})))| \tag{57}$$

$$+ |\sup_{f \in F}(\frac{1}{m}\sum_{i=1}^{m} f(g(x_t^i, \beta_0)) - \frac{1}{n}\sum_{i=1}^{n} f(h(x_s^i, \lambda_0))) - \sup_{f \in F}(E_{x_t}f(g(x_t, \beta_0)) - E_{x_s^i}f(h(x_s^i, \lambda_0)))| \tag{58}$$

We use the fact that $|\sup_x f_1(x) - \sup_x f_2(x)|$ is smaller than $\sup_x |f_1(x) - f_2(x)|$

$\hat{\lambda}, \lambda_0 \in \Omega_\lambda, \hat{\beta}, \beta_0 \in \Omega_\beta$ and get

$$\leq 2 \sup_{\lambda \in \Omega_\lambda, \beta \in \Omega_\beta} \sup_{f \in F} |(\frac{1}{m}\sum_{i=1}^{m} f(g(x_t^i, \beta)) - \frac{1}{n}\sum_{i=1}^{n} f(h(x_s^i, \lambda))) - (E_{x_t}f(g(x_t, \beta)) - E_{x_s^i}f(h(x_s^i, \lambda)))| \tag{59}$$

$$\leq 2 \sup_{\lambda \in \Omega_\lambda, \beta \in \Omega_\beta} \sup_{f \in F} |(\frac{1}{m}\sum_{i=1}^{m} f(g(x_t^i, \beta)) - E_{x_t}f(g(x_t, \beta))| \tag{60}$$

$$+ 2 \sup_{\lambda \in \Omega_\lambda, \beta \in \Omega_\beta} \sup_{f \in F} |(\frac{1}{n}\sum_{i=1}^{n} f(h(x_s^i, \lambda)) - E_{x_s^i}f(h(x_s^i, \lambda)))| \tag{61}$$

$$= 2 \sup_{\beta \in \Omega_\beta} \sup_{f \in F} |(\frac{1}{m}\sum_{i=1}^{m} f(g(x_t^i, \beta)) - E_{x_t}f(g(x_t, \beta))| + 2 \sup_{\lambda \in \Omega_\lambda} \sup_{f \in F} |(\frac{1}{n}\sum_{i=1}^{n} f(h(x_s^i, \lambda)) - E_{x_s^i}f(h(x_s^i, \lambda)))| \tag{62}$$

The results follows from Lemma 1.3, by noticing that for every random variable $W, Z$, constant $a, b$, we have $P(W + Z > a + b) \leq P(W > a) + P(Z > b)$.
Thus, for any $\alpha > 0$, with probability at least $1 - \alpha$.

$$\|E_{x_s^i}\mathcal{K}(h(x_s^i, \hat{\lambda}), .) - E_{x_t}\mathcal{K}(g(x_t, \hat{\beta})\|_H - \|E_{x_s^i}\mathcal{K}(h(x_s^i, \lambda_0), .) - E_{x_t}\mathcal{K}(g(x_t, \beta_0)\|_H \tag{63}$$

$$\leq 2\frac{\sqrt{K}}{\sqrt{n}}\left(4 + \sqrt{C^{(h,\alpha)} + \frac{d_\lambda}{2r_h}\log n}\right) + 2\frac{\sqrt{K}}{\sqrt{m}}\left(4 + \sqrt{C^{(g,\alpha)} + \frac{d_\beta}{2r_g}\log m}\right) \tag{64}$$

where $C^{(h,\alpha)} = \log(2|\Omega_\lambda|) + \log \alpha^{-1} + \frac{d_\lambda}{r_h} \log \frac{L_h}{\sqrt{K}}$, and $C^{(g,\alpha)} = \log(2|\Omega_\beta|) + \log \alpha^{-1} + \frac{d_\beta}{r_g} \log \frac{L_g}{\sqrt{K}}$

$\square$

**Lemma 4.2 (Consistency).** *Under* $\mathbf{H_0}$, *the estimators* $\hat{\lambda}$ *and* $\hat{\beta}$ *are consistent.*

*Proof.* We assume that $\Omega_\lambda, \Omega_\beta$ are bounded.

For notational convinence we simply call $||E_{x_s} \mathcal{K}(h(x_s, \hat{\lambda}), .) - E_{x_t} \mathcal{K}(g(x_t, \hat{\beta}), .)||_H - ||E_{x_s} \mathcal{K}(h(x_s, \lambda_0), .) - E_{x_t} \mathcal{K}(g(x_t, \beta_0), .)||_H$ as $\zeta(\hat{\lambda}, \hat{\beta})$.

Notice that $\zeta(\cdot)$ is continuous because $\zeta(\cdot)^2$ is the summation of expectations of bounded continuous functions (because the kernel is bounded continuous). If $(\hat{\lambda}, \hat{\beta})$ doesn't converge to $(\lambda_0, \beta_0)$ when $\zeta(\hat{\lambda}, \hat{\beta})$ converges to 0, then we have a sequence $(\hat{\lambda}_k, \hat{\beta}_k)$ and an $\epsilon > 0$, such that $||(\hat{\lambda}_k, \hat{\beta}_k) - (\lambda_0, \beta_0)|| > \epsilon$ but $\zeta(\hat{\lambda}_k, \hat{\beta}_k)$ converges to 0. Because $\Omega_\lambda, \Omega_\beta$ bounded, $\zeta(\cdot)$ is continuous, and hence $T(\lambda, \beta, C) = \{\lambda \in \Omega_\lambda, \beta \in \Omega_\beta | \zeta(\lambda, \beta) < C\}$ is a compact set of $(\lambda, \beta)$ for some constant C.

So we can find a point $(\tilde{\lambda}, \tilde{\beta})$ in $T(\lambda, \beta, C) \cap \{(\lambda, \beta) | ||(\lambda, \beta) - (\lambda_0, \beta_0)|| > \epsilon\}$ such that there is a subsequence $(\hat{\lambda}_{k_l}, \hat{\beta}_{k_l})$ which converges to $(\tilde{\lambda}, \tilde{\beta})$ when $l$ goes to $\infty$, based on Bolzano-Weierstrass theorem. But since $\zeta(\cdot)$ is continuous, we have $\zeta(\tilde{\lambda}, \tilde{\beta}) = 0$, with $||(\tilde{\lambda}, \tilde{\beta}) - (\lambda_0, \beta_0)|| > \epsilon$. This contradicts with the unique solution requirement of $(\lambda_0, \beta_0)$.

$\square$

**Theorem 4.3 (Hypothesis Testing).** *(a) Whenever* $\mathbf{H_0}$ *is true, with probability at least* $1 - \alpha$,

$$0 \le \mathcal{M}(\hat{\lambda}, \hat{\beta}) \le \sqrt{\frac{2K(m+n)\log \alpha^{-1}}{mn}} + \frac{2\sqrt{K}}{\sqrt{n}} + \frac{2\sqrt{K}}{\sqrt{m}} \tag{65}$$

*(b) Whenever* $\mathbf{H_A}$ *is true, with probability at least* $1 - \epsilon$,

$$\mathcal{M}(\hat{\lambda}, \hat{\beta}) \le \mathcal{M}^*(\lambda_A, \beta_A) + \sqrt{\frac{2K(m+n)\log \epsilon^{-1}}{mn}} + \frac{2\sqrt{K}}{\sqrt{n}} + \frac{2\sqrt{K}}{\sqrt{m}}$$

$$\mathcal{M}(\hat{\lambda}, \hat{\beta}) \ge \mathcal{M}^*(\lambda_A, \beta_A) - \frac{\sqrt{K}}{\sqrt{n}}\left(4 + \sqrt{C^{(h,\epsilon)} + \frac{d_\lambda}{2r_h}\log n}\right) - \frac{\sqrt{K}}{\sqrt{m}}\left(4 + \sqrt{C^{(g,\epsilon)} + \frac{d_\beta}{2r_g}\log m}\right)$$

$$\tag{66}$$

*where* $C^{(h,\epsilon)} = \log(2|\Omega_\lambda|) + \log \epsilon^{-1} + \frac{d_\lambda}{r_h} \log \frac{L_h}{\sqrt{K}}$, *and* $C^{(g,\epsilon)} = \log(2|\Omega_\beta|) + \log \epsilon^{-1} + \frac{d_\beta}{r_g} \log \frac{L_g}{\sqrt{K}}$

*Proof.* Under $\mathbf{H_0}$,

$$\mathcal{M}(\hat{\lambda}, \hat{\beta}) - \mathcal{M}^*(\lambda_0, \beta_0) \tag{67}$$
$$\le \mathcal{M}(\lambda_0, \beta_0) - \mathcal{M}^*(\lambda_0, \beta_0) \tag{68}$$
$$= MMD(h(x_s, \lambda_0), g(x_t, \beta_0)) - MMD^*(h(x_s, \lambda_0), g(x_t, \beta_0)) \tag{69}$$

where $MMD^*(\cdot)$ is the MMD in the population sense while $MMD(\cdot)$ takes the expectation in a sample sense. The MMD empirical bound from Theorem 7 in [1] can be directly applied to the right hand side of the above inequality. This application will lead to the bound on $\mathbf{H_0}$ in (65).

Similarly, under $\mathbf{H_A}$,

$$\mathcal{M}(\hat{\lambda}, \hat{\beta}) - \mathcal{M}^*(\lambda_A, \beta_A) \tag{70}$$
$$\le \mathcal{M}(\lambda_A, \beta_A) - \mathcal{M}^*(\lambda_A, \beta_A) \tag{71}$$
$$= MMD(h(x_s, \lambda_A), g(x_t, \beta_A)) - MMD^*(h(x_s, \lambda_A), g(x_t, \beta_A)) \tag{72}$$

Similar to the case of $\mathbf{H_0}$, the upper bound follows from Theorem 7 of [1]. The lower bound proof under the alternative follows from Lemma 1.3.

$$|\mathcal{M}(\hat{\lambda}, \hat{\beta}) - \mathcal{M}^*(\lambda_A, \beta_A)| = |\min_{\lambda \in \Omega_\lambda} \min_{\beta \in \Omega_\beta} \sup_{f \in F}(\frac{1}{m}\sum_{i=1}^m f(g(x_t^i, \beta)) - \frac{1}{n}\sum_{i=1}^n f(h(x_s^i, \lambda))) \quad (73)$$

$$- \min_{\lambda \in \Omega_\lambda} \min_{\beta \in \Omega_\beta} \sup_{f \in F}(E_{x_t} f(g(x_t, \beta)) - E_{x_s} f(h(x_s, \lambda)))| \quad (74)$$

Use the fact that $|\min_x f_1(x) - \min_x f_2(x)|$ is smaller than $\sup_x |f_1(x) - f_2(x)|$, we have

$$|\mathcal{M}(\hat{\lambda}, \hat{\beta}) - \mathcal{M}^*(\lambda_A, \beta_A)| \quad (75)$$

$$\leq \sup_{\lambda \in \Omega_\lambda, \beta \in \Omega_\beta} \sup_{f \in F} |(\frac{1}{m}\sum_{i=1}^m f(g(x_t^i, \beta)) - \frac{1}{n}\sum_{i=1}^n f(h(x_s^i, \lambda))) - (E_{x_t} f(g(x_t, \beta)) - E_{x_s} f(h(x_s, \lambda)))|$$

$$(76)$$

$$\leq \sup_{\beta \in \Omega_\beta} \sup_{f \in F} |\frac{1}{m}\sum_{i=1}^m f(g(x_t^i, \beta)) - E_{x_t} f(g(x_t, \beta))| + \sup_{\lambda \in \Omega_\lambda} \sup_{f \in F} |\frac{1}{n}\sum_{i=1}^n f(h(x_s^i, \lambda)) - E_{x_s} f(h(x_s, \lambda))|$$

$$(77)$$

The results come from Lemma 1.3, and the fact that for every random variable $W, Z$, constant $a, b$, we have $P(W + Z > a + b) \leq P(W > a) + P(Z > b)$ $\hfill\square$

**Lemma 4.4 (Linear transformation).** *Under* $\mathbf{H_0}$, *identity* $g(\cdot)$ *with* $h = \phi(x_s)^T \lambda$, *we have* $\Omega_\lambda := \{\lambda; |\frac{1}{n}\sum_{i=1}^n \|x_t^i - \phi(x_s^i)^T \lambda\|^2 \leq 3\sum_{k=1}^p Var(x_{t,k}) + \epsilon\}$. *For any* $\epsilon, \alpha > 0$ *and sufficiently large sample size, a neighborhood of* $\lambda_0$ *is contained in* $\Omega_\lambda$ *with probability at least* $1 - \alpha$.

*Proof.* Let $\phi(x_s) = (\phi_1, \phi_2, \ldots, \phi_p)$, and define $\Sigma_k = Var(\phi_k)$, $\mu_k = \mathbb{E}(\phi_k)$. Then we have

$$E[\|x_t - \phi(x_s)'\lambda\|^2] = E[\|\phi(\widetilde{x_s})'\lambda_0 - \phi(x_s)'\lambda\|^2] \quad (78)$$

$$= \lambda_0' \sum_{k=1}^p \Sigma_k \lambda_0 + \lambda' \sum_{k=1}^p \Sigma_k \lambda + (\lambda - \lambda_0)' \sum_{k=1}^p \mu_k \mu_k'(\lambda - \lambda_0) \quad (79)$$

$$\leq 3\lambda_0' \sum_{k=1}^p \Sigma_k \lambda_0 + (\lambda - \lambda_0)' \sum_{k=1}^p (2\Sigma_k + \mu_k \mu_k')(\lambda - \lambda_0) \quad (80)$$

$$= 3 \sum_{k=1}^p Var(x_{t,k}) + (\lambda - \lambda_0)' \sum_{k=1}^p (2\Sigma_k + \mu_k \mu_k')(\lambda - \lambda_0) \quad (81)$$

The set $S_2 = \{\lambda | E[\|x_t - \phi(x_s)'\lambda\|^2] \leq 3\lambda_0' \sum_{k=1}^p \Sigma_k \lambda_0 + C\}$ includes the set $S_1 = \{\lambda | (\lambda - \lambda_0)' \sum_{k=1}^p (2\Sigma_k + \mu_k \mu_k')(\lambda - \lambda_0) \leq C\}$ for any $C \geq 0$.
$S_1$ is a neighbourhood of $\lambda_0$ as an eclipse characterized $\sum_{k=1}^p (2\Sigma_k + \mu_k \mu_k')$. Now we see that our proposed trust region is $S_2$.
Further, we can just set $\epsilon = 0$ since whenever $\lambda = \lambda_0$, the upper bound should be exactly $2\lambda_0' \sum_{k=1}^p \Sigma_k \lambda_0$ instead of $3\lambda_0' \sum_{k=1}^p \Sigma_k \lambda_0$, which means there already exists some relaxations. $\hfill\square$

## 4.1 Additional results pertaining to the constants

**Lemma 4.5.** *Let* $S(\lambda, C) = \{\lambda \in \Omega_\lambda | \exists \beta \in \Omega_\beta \text{ s.t. } \|E_{x_t}\mathcal{K}(g(x_t, \beta).) - E_{x_s}\mathcal{K}(h(x_s, \lambda), .)\|_H - \|E_{x_t}\mathcal{K}(g(x_t, \beta_A).) - E_{x_s}\mathcal{K}(h(x_s, \lambda_A), .)\|_H \leq C\}$. *Under alternative, we could replace* $|\Omega_\lambda|$ *in* $C^{(h,\alpha)}$ *from (66) by* $|S(\lambda, C)|$ *for any* $C > 0$ *whenever* $m$ *and* $n$ *are large enough, and theorem 4.1 still holds. The result applies to* $\beta$ *as well.*

*Proof.*

$$\mathcal{M}^*(\lambda_A, \beta_A) - \mathcal{M}(\hat{\lambda}, \hat{\beta}) \tag{82}$$

$$= \sup_{f \in F}(E_{x_t}f(g(x_t, \beta_A)) - E_{x_s}f(h(x_s, \lambda_A))) - \min_{\lambda \in \Omega_\lambda} \min_{\beta \in \Omega_\beta} \sup_{f \in F}(\frac{1}{m}\sum_{i=1}^{m} f(g(x_t^i, \beta)) - \frac{1}{n}\sum_{i=1}^{n} f(h(x_s^i, \lambda))) \tag{83}$$

$$= \sup_{\lambda \in \Omega_\lambda} \sup_{\beta \in \Omega_\beta} [\sup_{f \in F}(E_{x_t}f(g(x_t, \beta_A)) - E_{x_s}f(h(x_s, \lambda_A))) - \sup_{f \in F}(\frac{1}{m}\sum_{i=1}^{m} f(g(x_t^i, \beta)) - \frac{1}{n}\sum_{i=1}^{n} f(h(x_s^i, \lambda)))] \tag{84}$$

$$= \sup_{\lambda \in \Omega_\lambda} \sup_{\beta \in \Omega_\beta} \{[\sup_{f \in F}(E_{x_t}f(g(x_t, \beta_A)) - E_{x_s}f(h(x_s, \lambda_A))) - \sup_{f \in F}(E_{x_t}f(g(x_t, \beta)) - E_{x_s}f(h(x_s, \lambda)))] \tag{85}$$

$$+ [\sup_{f \in F}(E_{x_t}f(g(x_t, \beta)) - E_{x_s}f(h(x_s, \lambda))) - \sup_{f \in F}(\frac{1}{m}\sum_{i=1}^{m} f(g(x_t^i, \beta)) - \frac{1}{n}\sum_{i=1}^{n} f(h(x_s^i, \lambda)))]\} \tag{86}$$

Similar to $S(\lambda, C)$, let $T(\beta, C) = \{\beta \in \Omega_\beta | \exists \lambda \in \Omega_\lambda \text{ s.t. } ||E_{x_t}\mathcal{K}(g(x_t, \beta).) - E_{x_s}\mathcal{K}(h(x_s, \lambda), .)||_H - ||E_{x_t}\mathcal{K}(g(x_t, \beta_A).) - E_{x_s}\mathcal{K}(h(x_s, \lambda_A), .)||_H \leq C\}$.

For any $\epsilon > 0$, $\sup_{\lambda \in \Omega_\lambda, \beta \in \Omega_\beta}(\cdot) = \sup_{((\lambda, \beta) \in S(\lambda, C) \times T(\beta, C), (\lambda, \beta) \notin S(\lambda, C) \times T(\beta, C))}(\cdot)$.

$(\lambda_A, \beta_A)$ is contained in the $S(\lambda, C) \times T(\beta, C)$. So simply using $(\lambda, \beta) = (\lambda_A, \beta_A)$ will give us a lower bound on the corresponding set. Hence

$$0 + [\sup_{f \in F}(E_{x_t}f(g(x_t, \beta_A)) - E_{x_s}f(h(x_s, \lambda_A))) - \sup_{f \in F}(\frac{1}{m}\sum_{i=1}^{m} f(g(x_t^i, \beta_A)) - \frac{1}{n}\sum_{i=1}^{n} f(h(x_s^i, \lambda_A)))] \tag{87}$$

$$\geq 0 - \sqrt{2K}\sqrt{\frac{m+n}{mn}}\sqrt{\log \epsilon^{-1}} - \frac{2\sqrt{K}}{\sqrt{n}} - \frac{2\sqrt{K}}{\sqrt{m}} \tag{88}$$

with probability at least $1 - \epsilon$. The last inequality follow from Theorem 7 in [1].

Alternatively when $(\lambda, \beta) \notin S(\lambda, C) \times T(\beta, C)$, the first term $[\sup_{f \in F}(E_{x_t}f(g(x_t, \beta_A)) - E_{x_s}f(h(x_s, \lambda_A))) - \sup_{f \in F}(E_{x_t}f(g(x_t, \beta)) - E_{x_s}f(h(x_s, \lambda)))]$ is uniformly smaller than -C, which follow from condition in Lemma and Riesz Representation theorem.

The second term $\sup_{\lambda \in \Omega_\lambda} \sup_{\beta \in \Omega_\beta} [\sup_{f \in F}(E_{x_t}f(g(x_t, \beta)) - E_{x_s}f(h(x_s, \lambda))) - \sup_{f \in F}(\frac{1}{m}\sum_{i=1}^{m} f(g(x_t^i, \beta)) - \frac{1}{n}\sum_{i=1}^{n} f(h(x_s^i, \lambda)))]$ appeared before when we proved lower bound under alternative. So we use these results given in the proof procedure of Theorem 4.3, we have the following with probability greater than $1 - \epsilon$.

The second term is smaller than $-C + \frac{\sqrt{K}}{\sqrt{n}}\left(4 + \sqrt{C^{(h,\epsilon)} + \frac{d_\lambda}{2r_h}\log n}\right) + \frac{\sqrt{K}}{\sqrt{m}}\left(4 + \sqrt{C^{(g,\epsilon)} + \frac{d_\beta}{2r_g}\log m}\right)$.

So any C, which satisfies $0 - \sqrt{2K}\sqrt{\frac{m+n}{mn}}\sqrt{\log \epsilon^{-1}} - \frac{2\sqrt{K}}{\sqrt{n}} - \frac{2\sqrt{K}}{\sqrt{m}} > -C + \frac{\sqrt{K}}{\sqrt{n}}\left(4 + \sqrt{C^{(h,\epsilon)} + \frac{d_\lambda}{2r_h}\log n}\right) + \frac{\sqrt{K}}{\sqrt{m}}\left(4 + \sqrt{C^{(g,\epsilon)} + \frac{d_\beta}{2r_g}\log m}\right)$, can then be used for computing the sets $S(\lambda, C), T(\beta, C)$.

The reason for this follows from the fact that the upper bound for second term would be smaller than lower bound for first term, which implies $\sup_{(\lambda, \beta) \in \Omega_\lambda \times \Omega_\beta}(\cdot)$ in (84) could be reduced to $\sup_{(\lambda, \beta) \in S(\lambda, C) \times T(\beta, C)}(\cdot)$. We point out that the proof procedure of the consistency lower bound under $\mathbf{H_A}$ ((66) from Theorem 4.3) also derives from (84). $\qquad \square$

# 5   Further Experimental Results

Figure 1: The histograms of minimal MMD statistics for three different source/target combinations – Normal vs. Normal, Normal vs. Laplace and Normal vs. Exponential. The six plots correspond to increasing sample sizes from $2^5$ to $2^{10}$.

Fig 1(c)(f) correlates to Fig 1 (e)(f) in the main body. The extra four sub-plots make up figures for other sample sizes from $2^5$ to $2^{10}$.

Figure 2:  Estimation error for the setting where source and target samples are three-dimensional. The six lines correspond to two different sets of experiments.

Section 7 includes model details.

Figure 3: Scatter plots of corrected Batch1 and Batch2 ($x$ and $y$ axes) measures for the 12 proteins using gold standard linear model (blue) and our minimal MMD based correction (red). The setting is $S_1$ where in only those samples whose corresponding source (Batch1) and target (Batch2) were available. Note that our model does not use this pairwise correspondence information unlike the gold standard.

The points in the scatter plot related to same persons appearing both in Batch 1 and Batch 2. Since that, we'd like to see those points lie on the diagonal line after transformed. It shows that our minimal MMD method works almost as good as gold standard method. The setting is $S_1$ where in only those samples whose corresponding source (Batch1) and target (Batch2) were available. Note that our model does not use this pairwise correspondence information unlike the gold standard.

Figure 4: Scatter plots of corrected Batch1 and Batch2 ($x$ and $y$ axes) measures for the 12 proteins using gold standard linear model (blue) and our minimal MMD based correction (red). The setting is $S_2$ where our model uses all samples – even those whose corresponding source (Batch1) and target (Batch2) were available. The gold standard plots (blue) nevertheless can only use pairwise samples (i.e., $S_1$, refer previous plot). Note that our model does not use this pairwise correspondence information unlike the gold standard.

The points in the scatter plot related to same persons appearing both in Batch 1 and Batch 2. Since that, we'd like to see those points lie on the diagonal line after transformed. It shows that our minimal MMD method works almost as good as gold standard method. The setting is $S_2$ where our model uses all samples – even those whose corresponding source (Batch1) and target (Batch2) were available. Note that our model does not use this pairwise correspondence information unlike the gold standard.

# 6 Illustration Figure for Comparison with MMD

Figure 5: Two normal distributions with different mean

MMD rejects the null hypothesis since two distributions are different. Our minimal MMD method accepts our null hypothesis, since two distributions can be matched after a simple transformation applied. Under this scenario, our method offers information and provides a transformation to match the two distributions, while MMD tells nothing more than they are different.

# 7 Simulation Model Details

Here are model details about Fig 1(c)(d) in the main body.
Fig 1 (c), model:

$$x_t \sim N(10, 4) \tag{89}$$
$$x_s \sim N(0, 1) \tag{90}$$
$$Model\ is\ x_t = a \times x_s + b \tag{91}$$

We generate samples from $N(10, 4)$ for $x_t$ and from $N(0, 1)$ for $x_s$. Then we fit model $x_t = a \times x_s + b$ using our minimal MMD. In other words, $h(x_s, \lambda) = a \times x_s + b$. The parameter $\lambda = (a, b)$. We call parameter 'a' slope and 'b' intercept. The $L_1$ error is $|a - 2|$ for slope curve and $|b - 10|$ for intercept curve. With sample sizes from $2^4$ to $2^{10}$, we get the curve.

Fig 1 (d), Model 1:

$$w \sim N\left(\begin{pmatrix} 0 \\ 0 \end{pmatrix}, \begin{pmatrix} 1 & 0.5 \\ 0.5 & 1 \end{pmatrix}\right) \tag{92}$$

$$x_s \sim N\left(\begin{pmatrix} 0 \\ 0 \end{pmatrix}, \begin{pmatrix} 1 & 0.5 \\ 0.5 & 1 \end{pmatrix}\right) \tag{93}$$

$$x_t = \begin{pmatrix} 1 & 2 & 10 \\ 2 & 1 & -20 \end{pmatrix} \begin{pmatrix} w \\ 1 \end{pmatrix} \tag{94}$$

$$Model\ is\ x_t = \begin{pmatrix} a_{11} & a_{12} & a_{13} \\ a_{21} & a_{22} & a_{23} \end{pmatrix} \begin{pmatrix} x_s \\ 1 \end{pmatrix} \tag{95}$$

We generate samples for $x_s$, $w$, then we transform w to get $x_t$. Based on samples, we fit a model

$$x_t = \begin{pmatrix} a_{11} & a_{12} & a_{13} \\ a_{21} & a_{22} & a_{23} \end{pmatrix} \begin{pmatrix} x_s \\ 1 \end{pmatrix} \tag{96}$$

Quartic Mean of estimation error for "Model 1, first row" is

$$\sqrt{\frac{(a_{11} - 1)^2 + (a_{12} - 2)^2 + (a_{13} - 10)^2}{3}} \tag{97}$$

Quartic Mean of estimation error for "Model 1, second row" is

$$\sqrt{\frac{(a_{21} - 2)^2 + (a_{22} - 1)^2 + (a_{23} + 20)^2}{3}} \tag{98}$$

Fig 1 (d), Model 2:

$$w \sim \begin{pmatrix} N(0, 1) \\ \chi_1^2 \end{pmatrix} \tag{99}$$

$$x_s \sim \begin{pmatrix} N(0, 1) \\ \chi_1^2 \end{pmatrix} \tag{100}$$

$$x_t = \begin{pmatrix} 1 & 2 & 10 \\ 2 & 1 & -20 \end{pmatrix} \begin{pmatrix} w \\ 1 \end{pmatrix} \tag{101}$$

$$Model\ is\ x_t = \begin{pmatrix} a_{11} & a_{12} & a_{13} \\ a_{21} & a_{22} & a_{23} \end{pmatrix} \begin{pmatrix} x_s \\ 1 \end{pmatrix} \tag{102}$$

We generate samples for $x_s$, $w$, then we transform w to get $x_t$. Based on samples, we fit a model

$$x_t = \begin{pmatrix} a_{11} & a_{12} & a_{13} \\ a_{21} & a_{22} & a_{23} \end{pmatrix} \begin{pmatrix} x_s \\ 1 \end{pmatrix} \tag{103}$$

Quartic Mean of estimation error for "Model 2, first row" is

$$\sqrt{\frac{(a_{11}-1)^2+(a_{12}-2)^2+(a_{13}-10)^2}{3}} \tag{104}$$

Quartic Mean of estimation error for "Model 2, second row" is

$$\sqrt{\frac{(a_{21}-2)^2+(a_{22}-1)^2+(a_{23}+20)^2}{3}} \tag{105}$$

Figure 2 in supplement, Model 1:

$$w \sim N\left(\begin{pmatrix} 0 \\ 0 \\ 0 \end{pmatrix}, \begin{pmatrix} 1 & 0.5 & 0.5 \\ 0.5 & 1 & 0.5 \\ 0.5 & 0.5 & 1 \end{pmatrix}\right) \tag{106}$$

$$x_s \sim N\left(\begin{pmatrix} 0 \\ 0 \\ 0 \end{pmatrix}, \begin{pmatrix} 1 & 0.5 & 0.5 \\ 0.5 & 1 & 0.5 \\ 0.5 & 0.5 & 1 \end{pmatrix}\right) \tag{107}$$

$$x_t = \begin{pmatrix} 1 & 0 & 2 & 10 \\ 1 & 2 & 0 & 20 \\ 0 & 2 & 1 & -30 \end{pmatrix} \begin{pmatrix} w \\ 1 \end{pmatrix} \tag{108}$$

$$Model\ is\ x_t = \begin{pmatrix} a_{11} & a_{12} & a_{13} & a_{14} \\ a_{21} & a_{22} & a_{23} & a_{24} \\ a_{31} & a_{32} & a_{33} & a_{34} \end{pmatrix} \begin{pmatrix} x_s \\ 1 \end{pmatrix} \tag{109}$$

We generate samples for $x_s$, $w$, then we transform w to get $x_t$. Based on samples, we fit a model

$$x_t = \begin{pmatrix} a_{11} & a_{12} & a_{13} & a_{14} \\ a_{21} & a_{22} & a_{23} & a_{24} \\ a_{31} & a_{32} & a_{33} & a_{34} \end{pmatrix} \begin{pmatrix} x_s \\ 1 \end{pmatrix} \tag{110}$$

Quartic Mean of estimation error for "Model 2, first row" is

$$\sqrt{\frac{(a_{11}-1)^2+(a_{12}-0)^2+(a_{13}-2)^2+(a_{14}-10)^2}{4}} \tag{111}$$

Quartic Mean of estimation error for "Model 2, second row" is

$$\sqrt{\frac{(a_{21}-1)^2+(a_{22}-2)^2+(a_{23}-0)^2+(a_{24}-20)}{4}} \tag{112}$$

Quartic Mean of estimation error for "Model 2, third row" is

$$\sqrt{\frac{(a_{31}-0)^2+(a_{32}-2)^2+(a_{33}-1)^2+(a_{34}+30)}{4}} \tag{113}$$

## References

[1] Arthur Gretton, Karsten M Borgwardt, Malte J Rasch, Bernhard Schölkopf, and Alexander Smola. A kernel two-sample test. *The Journal of Machine Learning Research*, 13(1):723–773, 2012.