[Reviews · NeurIPS 2016]

Reviewer 1

Summary

The article proposes a statistical procedure for unsupervised domain adaptation. The procedure is two-step. In the first, optimal transformation parameters are determined that minimizes MMD distance between the distributions of the transformed samples in both target and source domain. In the second step, the procedure tests the goodness of fit by computing the minimal MMD statistic of the adaptation and assigns a significance level. The article proves the consistency of the procedure as well as determine the power of the hypothesis test.

Qualitative Assessment

The article seems to have a high level of technical content. It extends earlier works proposed in [1] and [15] to more generic settings. The consistency proofs and the power analysis are in my opinion important technical results for the adaptation of the proposed procedure in practice. The experiments support the technical developments in the article. Especially, the synthetic data is welcomed. The real experiments have some missing details, which diminishes its impact. In the real data experiments, I think more details would have been more appropriate. The points that are not clear are: - As far as I can understand from the text, the linear transformation between the batches are determined using the corresponding samples. Does figure 2 show the errors in the other part of the data? - What are G, G’ and K used in the experiments? If the setting is the same as in Section 4, then the proposed model also looks for the best linear transformation. In this case, do we really expect a difference between the “ground truth” and the proposed model? - The real data experiments do not necessarily demonstrate the advantage of the proposed model. I suggest authors to mix different datasets to show the statistical power. For instance, if they were to use another data, i.e. not related to CSF proteins, with the same dimension can the statistical test figure out that they are not from the same distribution while the two batches are? Lastly, I do not believe the title is really appropriate. Application to neurodegenerative diseases is not necessarily a domain of neuroscience. Perhaps one can consider this in clinical neuroscience. In any case, I suggest authors to change the title to …Applications in Neurodegenerative Disease or …Applications in Clinical Neuroscience.

Confidence in this Review

2-Confident (read it all; understood it all reasonably well)


Reviewer 2

Summary

This paper introduces a new method for domain adaptation that relies the Maximum Mean Discrepancy (MMD). Transformations of both the source and target distributions are learned while jointly minimizing the MMD between the transformed versions. This yields a statistical test of whether the predefined function class is rich enough to have transformations that make the two distributions indistinguishable, and this test is shown to be consistent. The method is illustrated on batches of data from an Alzheimer's disease study, showing that shifting the distribution achieves some improvements in analytical performance.

Qualitative Assessment

The paper presents an interesting and smart way of performing covariate shift by aiming to make the two distributions indistinguishable by minimizing MMD. The paper however could benefit of more clarity and completeness so it can make impact. In terms of the applicability of this approach, the authors talk about the importance of being able to perform statistical tests and not just optimize the performance of a classifier. The bounds that they derive for their statistical test is useful to know how big the sample size should be to perform an appropriate shift. However, this doesn't say much on how it affects the scientific questions asked in the experiment, which need different statistical test. From the two sites where the data originates, we might (for example) have two types of patients. How does this shift help us with identifying them? Table 1 shows the answer to a similar question where the hippocampal volume is better predicted by the morphed CSF data, but more tests such as this one are required to believe that the method is useful. What if the question of interest was to find the CSF proteins that are most predictive, how would the shifting of the covariates affect our ability to answer the question? More experiments, as well as a better explanation of the problem would make the paper stronger. For example, what if the data from the two sites were unevenly distributed such as they had different proportions of the possible phenotypes. The variance we would see would not be a consequence of the measurement only, but also reflects population variance. The proposed measure could however homogenize the two batches and introduce some error. Perhaps these questions are hard to answer theoretically but more experimental results could help. It would be interesting to also mention how this method could be applied to more than two batches.

Confidence in this Review

2-Confident (read it all; understood it all reasonably well)


Reviewer 3

Summary

This paper propose to perform a statistical test checking of the domain adaptation algorithm using ideas from hypothesis testing. The proposed algorithm is analyzed in terms of convergence, consistency, and lower-bound deduction. All in all, this paper is good in self-contained.

Qualitative Assessment

This paper spend too much space to explain the background and relevant works. However, due to the abundancy contents, each part is explained too brief, which makes it hard to understand, especially the experimental section. Indeed, I have to check the supplementary files often to help understanding.

Confidence in this Review

2-Confident (read it all; understood it all reasonably well)


Reviewer 4

Summary

The authors propose an extension of the DA approach presented in [1] where they generalize the linear feature-transforamtions and also define a hypothesis test to assure that the target transformations can successfully minimize the domain mismatch based on MMD.

Qualitative Assessment

As mentioned above, the paper has a very nice theprethical formulation (that generalizes the work in [1]). Also, using the hypothesis testing as an extra check-up step is novel. I only don't see why the authors haven't compared this method with related works on DA as there are plenty (also in their related work...)/

Confidence in this Review

3-Expert (read the paper in detail, know the area, quite certain of my opinion)


Reviewer 5

Summary

This paper discusses domain adaptation to match the distribution of two sources as measured by MMD. A parameterized transformation is applied to one or both sources to minimize the MMD objective, and the outcome of the optimization is used to conduct a hypothesis test in order to decide if the parameterized family is capable of matching the two sources. The authors prove consistency and convergence results for their method, provide confidence bounds for hypothesis testing and apply recent advances in non-convex optimization to obtain subsequently tighter bounds on the optimization objective.

Qualitative Assessment

The idea of applying transformations to minimize MMD between two sources seems new and interesting, but rather straight forward given previous work on MMD hypothesis testing. The main contribution consists of several sophisticated technical results and optimization strategies. These are valuable results but should be explained in greater detail. Intuitive explanation of the theorems could be helpful for readers who haven't looked at the proofs. Plots or illustrations can also be of great help. Some of the important theorems, such as Theorem4.3, can be easier to read if the notation and symbols used are more clearly explained.

Confidence in this Review

2-Confident (read it all; understood it all reasonably well)


Reviewer 6

Summary

The authors present an approach to perform kernel-based null-hypothesis testing for two distributions from different sources. The text is well written and includes both theoretical and empirical simulation results that appear to back up the conclusions.

Qualitative Assessment

Overall, it is not clear what real-world cross-validation scenarios actually really require or profit from the proposed approach in the domain of neuroscience. However, the paper is rather distant from the reviewer's main areas of expertise. Here are few concrete concerns: a) It is unclear why the impact on inference between different source and target distributions is introduced by an *unsupervised* setting. b) Distinguishing between "formulation" and "statistical" aspects of the problem at hand. c) typo: "functions classes"

Confidence in this Review

2-Confident (read it all; understood it all reasonably well)